# Functionalized Large-Pore Mesoporous Silica Microparticles for Gefitinib and Doxorubicin Codelivery

**DOI:** 10.3390/ma12050766

**Published:** 2019-03-06

**Authors:** Yan Li, Fangxiang Song, Liang Cheng, Jin Qian, Qianlin Chen

**Affiliations:** 1Institute of Advanced Technology, Guizhou University, Guiyang 550025, China; yanli@gzu.edu.cn; 2School of Chemistry and Chemical Engineering, Guizhou University, Guiyang 550025, China; sfxgzh1902@163.com; 3School of Electrical Engineering, Guizhou University, Guiyang 550025, China; lcheng1@gzu.edu.cn (L.C.); jqian@gzu.edu.cn (J.Q.)

**Keywords:** mesoporous silica, multiamine chains, carboxylation, interfacial functionalization, great synergistic effects

## Abstract

Large-pore coralline mesoporous silica microparticles (CMS) were synthesized using the triblock polymer PEG-*b*-PEO-*b*-PEG and a hydrothermal method. Scanning electron microscopy (SEM) and transmission electron microscopy (TEM) revealed the coralline morphology of the fabricated materials. The Brunauer–Emmett–Teller (BET) method and the Barrett–Joyner–Halenda (BJH) model confirmed the existence of large pores (20 nm) and of a tremendous specific surface area (663.865 m^2^·g^−1^) and pore volume (0.365 cm^3^·g^−1^). A novel pH-sensitive multiamine-chain carboxyl-functionalized coralline mesoporous silica material (CMS–(NH)_3_–COOH) was obtained via a facile “grafting-to” approach. X-ray photoelectron spectroscopy (XPS) and Fourier-transform infrared spectroscopy (FT-IR) validated the effective interfacial functionalization of CMS with carboxyl and multiamine chains. The encapsulation and release behavior of the dual drug (gefitinib (GB) and doxorubicin (DOX)) was also investigated. It was found that CMS–(NH)_3_–COOH allows rapid encapsulation with a high loading capacity of 47.36% for GB and 26.74% for DOX. Furthermore, the release profiles reveal that CMS–(NH)_3_–COOH can preferably control the release of DOX and GB. The accumulative release rates of DOX and GB were 32.03% and 13.66%, respectively, at a low pH (pH 5.0), while they reduced to 8.45% and 4.83% at pH 7.4. Moreover, all of the modified silica nanoparticles exhibited a high biocompatibility with a low cytotoxicity. In particular, the cytotoxicity of both of these two drugs was remarkably reduced after being encapsulated. CMS–(NH)_3_–COOH@GB@DOX showed tremendously synergistic effects of the dual drug in the antiproliferation and apoptosis of A549 human cancer cells in vitro.

## 1. Introduction

Cancer is the leading cause of death in the world. In 2016, there were 17.2 million cancer cases worldwide and 8.9 million deaths [1]. Despite the risk of dying from cancer being reduced in all countries with reliable data, the mortality rates for individuals with lung cancer are increasing, which is a priority for prevention and further research [2]. Chemotherapy is the most common choice in cancer treatment, but it can cause many unwanted side effects. It is important to explore different strategies to enhance cancer therapy. Multidrug combination therapy has become a promising strategy for producing a synergistic response that enhances cancer chemotherapeutic responses [3,4,5,6,7,8]. The most recent pattern is the combination of traditional chemotherapy drugs and the tyrosine kinase inhibitor [9,10,11,12]. There are a number of carriers, such as nanoparticles, thermosensitive hydrogels, micelles, and complexed nanoparticles [13,14,15,16,17,18].

Among various anticancer drugs, water-soluble doxorubicin hydrochloride (doxorubicin hydrochloride) is an effective chemotherapy drug that has been approved for the treatment of various cancers. On the other hand, gefitinib is a specific inhibitor of the epidermal growth factor receptor (EGFR) tyrosine kinase and has been shown to suppress the activation of EGFR signaling required for cell survival and proliferation in non-small-cell lung-cancer (NSCLC) cell lines. It was reported that the codelivery of gefitinib and doxorubicin (DOX) can be used for targeted combination chemotherapy [9,19,20].

Researchers [6] have successfully synthesized a series of mesoporous silica materials in the early 1990s. In 2001, the MCM-41-type mesoporous silica material was first reported as a drug-delivery system. After more than 10 years of development, mesoporous silica materials are now widely used in the field of drug delivery [21,22,23,24] (the most commonly used drugs are ibuprofen (IBU) [25,26], aspirin [27], and DOX [28]). They have attracted much attention in the multidrug codelivery field because of their multitude of desirable properties, such as a large specific surface area and entrance, a good biocompatibility, and a low toxicity. AXT/CST-loaded composite nanoparticles (ACML) were prepared in mesoporous silica nanoparticles (MSN) and axtinib (AXT) in a polyethylene glycol liposome bilayer by Choi et al. ACML induces synergistic cancer-cell apoptosis compared to cocktails (AXT/CST). ACML inhibits cell proliferation by blocking mitochondrial function, induces apoptosis, and enhances antitumor effects [7]. The pH-sensitive lipid bilayer (HHG_2_C_18_-L) was constructed by using a synthetic zwitterionic oligopeptide lipid (1,5-dioctadecyl-_L_-glutamy12-histidyl-hexahydrobenzoic acid, HHG_2_C_18_) and applied to coat amino-functionalized mesoporous silica nanoparticles MSN–NH_2_. Erlotinib and DOX were added to HHG_2_C_18_-L and MSN–NH_2_, respectively, to obtain pH-sensitive charge-converted Erlotinib/DOX co-loaded nanoparticles (M-HHG_2_C_18_-L (E + D)). Compared with nonsensitive erlotinib/DOX codelivery nanoparticles (M-SPC–L (E + D)) and the concurrent drug combination, M-HHG_2_C_18_-L (E + D) has the properties of sequential cross-linked drug release and pH-sensitive charge transfer, and it has a significant synergistic effect on the antiproliferation and apoptosis of A549 human cancer cells in vitro [15].

Based on previous methods [29], we designed a facile synthetic route to produce carboxyl-functionalized large-pore mesoporous silica materials that are able to effectively encapsulate and release gefitinib and doxorubicin. To obtain the desired large pores, we employed triblock poly (ethylene glycol)-b-poly (propylene glycol)-b-(poly(ethylene glycol) (PEG-b-PPG-b-PEG) as the structure-directing template rather than the commonly used block copolymer (Pluronic P123 and F127). Transmission electron microscopy (TEM) images of the obtained materials (large-pore coralline mesoporous silica microparticles; CMS) show that the material possesses an unambiguous open-framework structure. This structure favors efficient mass transfer, as the high-density entrances enable rapid and efficient drug encapsulation. The functionalized part, because of carboxyl, can be combined with the silanol groups but cannot be directly carboxylic-functionalized on a silica mesoporous surface. As it cannot be directly carboxylic-functionalized on a silica mesoporous surface, we designed the amino-functionalized mesoporous silica microparticles by grafting and combined succinic anhydride with the amino to obtain carboxylic-functionalized materials with multiamine chains (CMS–(NH)_3_–COOH) (see Scheme 1).

The CMS and CMS–(NH)_3_–COOH were then investigated as drug-delivery vehicles. As shown in the release experiments in a weak-basic (pH 7.4) and weak-acid (pH 5.0) buffer solutions, they exhibited a high loading capacity and a sustained release of gefitinib and doxorubicin, which verified the significance of the designed experiment and the value of the research (the release rate of DOX and GB in the weak-acid solution was higher than in the weak-basic solution, manifesting the pH sensitivity of CMS–(NH)_3_–COOH).

We applied this novel pH-sensitive multiamine-chain carboxyl-functionalized coralline mesoporous silica carrier for A549 human cancer cells in vitro. Gefitinib and DOX were incorporated into CMS–(NH)_3_–COOH to obtain pH-sensitive carboxyl-functionalized large-pore mesoporous silica gefitinib/DOX codelivery microparticles (CMS–(NH)_3_–COOH@DOX@GB). Two drugs were conjugated to CMS–(NH)_3_–COOH and showed an enhanced uptake in A549 human cancer cells compared with the microparticles obtained with only gefitinib. IC50 doses revealed the potential antiproliferative effect on A549 human cancer cells. Therefore, the dual-drug conjugated CMS–(NH)_3_–COOH could be a potential drug carrier for active therapeutic aspects in cancer therapy.

## 2. Materials and Methods

### 2.1. Preparation of Carboxyl-Functionalized Large-Pore Mesoporous Silica Microparticles

Four grams of triblock poly (ethylene glycol)-*b*-poly (propylene glycol)-*b*-(poly(ethylene glycol) (PEG-*b*-PPG-*b*-PEG), MW = 8400, Aladdin, Shanghai, China) and 20 mL of hydrochloric acid (36–38%, Chongqing, China) were added into 120 mL deionized water. This was stirred for 20 min until the solution was clarified at room temperature; we then added a 6 g sodium silicate (Komeo, Tianjin, China) solution (10.5 g sodium silicate dissolved in 10.5 g deionized water). After stirring for 4 h, 15 mL of anhydrous ethanol was added dropwise to the rest of the 15 g sodium silicate solution, adding the mixed solution at a rate of 0.3 mL/s. Next, it was washed at a constant pressure in a split funnel with 5 mL deionized water; we obtained a pale-blue solution after stirring for 24 h at 1200 rpm. The above mixed solution was poured into the reaction kettle with the polytetrafluoroethylene gallbladder hydrothermal for 24 h at 120 °C with a heating rate of 5 °C/min. The mixture was obtained by a vacuum-suction filter, washed three times with deionized water and anhydrous ethanol, and dried overnight in a drum wind-drying oven at 60 °C. The obtained solid white powder was calcined at 6 h in a high-temperature tubular-resistant furnace of 550 °C on air with a temperature of 5 °C/min; this was named CMS.

We mixed 500 mg of CMS with 40 mL of dry toluene and performed ultrasonic dispersion for 30 min. Then, 1.5 mL 3-[2-(2-aminoethylamino) ethylamino]propyl-trimethoxysilane (AEPTMS, 95%, Macklin, Shanghai, China) was added and stirred for 12 h at 100 °C. The samples obtained by centrifugation, washed three times, and dried for 6 h at 60 °C were named CMS–(NH)_3_.

Then, 500 mg of CMS–(NH)_3_ was dispersed in a 50 mL pyridine solution and stirred for 1 h at 25 °C. Subsequently, 4.5 g of succinic anhydride (SAD, 99%, Aladdin, Shanghai, China) was added and stirred for 6 h at 110 °C. After cooling to room temperature, the carboxyl-functionalized materials were obtained by filtering, were washed three times with ethanol and deionized water, were dried at 35 °C for 24 h, and were named CMS–(NH)_3_–COOH.

### 2.2. Characterization Techniques

#### 2.2.1. Morphology and Measurements

Scanning electron microscopy (SEM) images were taken with a JEOL-JSM-7500F (Japan Electronics corporation, Tokyo, Japan) electron microscope operating at 20 KV. Transmission electron microscopy (TEM) images of samples were observed on a JEM-2100 (Japan Electronics Corporation, Tokyo, Japan) electron microscope at 220 KV.

#### 2.2.2. FT-IR Spectroscopy Measurements

Fourier-transform infrared (FT-IR) spectra were collected on a Nicolet iS50 (Thermo Fisher Scientific, San Diego, CA, USA) spectrophotometer using KBr pellets to analyze the bonding architecture and to identify the functional groups present in all samples.

#### 2.2.3. X-ray Diffraction (XRD) Analysis

The powder XRD patterns were recorded on a Bruker D8 X-ray diffractometer (Bruker Corporation, Karlsruhe, Germany) with Ni-filtered CuKα radiation (40 kV, 40 mA). The samples were scanned in the 2θ angular range of 5°–80° with a step size of 0.02° and a 2 s acquisition time/step.

#### 2.2.4. X-ray Photoelectron Spectroscopy (XPS) Analysis

XPS images of the samples were analyzed with an American Thermo Fisher K-Alpha (San Diego, CA, USA) at 12 KV to analyze the chemical environment of the elements.

#### 2.2.5. Nitrogen Adsorption–Desorption Analysis

Nitrogen sorption isotherms were measured with a Micrometrics ASAP 2020. Before measurement, the samples were degassed in a vacuum at 100 °C for at least 12 h. The Brunauer–Emmett–Teller (BET) method was utilized to calculate the specific surface areas (S_BET_) using the adsorption data. By using the Barrett–Joyner–Halenda (BJH) model, the pore volumes and pore-size distributions were derived from the adsorption branches of the isotherms.

#### 2.2.6. Zeta-Potential Analysis

Before the zeta potential of all samples was measured with a Zetasizer Nano ZS 90 (Malvern, UK), all samples were dispersed at different pH deionized water solutions (pH 5, 6, 7.4, and 8).

### 2.3. Drug-Loading Experiment

DOX (Aladdin, Shanghai, China) was loaded: 30 mg DOX was dissolved in 30 mL of 5% phosphate buffer saline (1.5 mL PBS, adding 28.5 mL, deionized), and 60 mg CMS–(NH)_3_–COOH was added. This was stirred at room temperature at 400 rpm for 24 h while in a dark condition. The loaded samples (named CMS–(NH)_3_–COOH@DOX) were collected after centrifugation at 8000 rpm for 10 min, were washed of the sediment to clarification with 5% PBS, and were dried in an air-dry oven at 50 °C for 24 h. To calculate the DOX loading rate [30], we used an ultraviolet spectrophotometer UV-6100s (Mapuda, Shanghai, China) at 480 nm and took the average after repeating three times to measure absorbance.

GB (Aladdin, Shanghai, China) was loaded: 30 mg GB was dissolved in 30 mL of dichloromethane (DCM), and 60 mg of CMS–(NH)_3_–COOH was added. This was stirred at room temperature at 400 rpm for 24 h in a dark condition. The loaded samples (named CMS–(NH)_3_–COOH@GB) were collected after being centrifuged at 8000 rpm for 10 min, were washed of the sediment to clarification with DCM, and were dried in an air-dry oven at 50 °C for 24 h. The following steps were the same as above except the absorbance was measured at 252 nm.

GB and DOX was co-loaded: 30 mg of GB was dissolved in 30 ml of dichloromethane (DCM), and 31 mg of CMS–(NH)_3_–COOH@DOX was added. The following steps were the same as above, and the loaded samples was named CMS–(NH)_3_–COOH@DOX@GB. To calculate the DOX and GB loading rate respectively, we used an ultraviolet spectrophotometer UV-6100s at 480 nm and 252 nm and took the average after repeating three times to measure the absorbance respectively.

### 2.4. In Vitro Cytotoxicity Experiment

For the in vitro cell culture, human non-small-cell lung-cancer cells (A549) were purchased from the Cell Bank of the Chinese Academy of Sciences (Shanghai, China). Cell culture: Normal A549 cells were cultured using a Dulbecco’s Modified Eagle’s medium (DMEM) culture medium supplemented with 10% (v/v) fetal bovine serum (FBS), antibiotic penicillin (100 U·mL^−1^), and streptomycin (100 μg·mL^−1^). The cells were cultivated under a humidified atmosphere with 5% CO_2_ at 37 °C; 3-(4,5-Dimethylthiazol-2-yl)-2,5-diphenylte-trazolium bromide (MTT) was purchased from Thermo Fisher Scientific.

The in vitro cytotoxicity of CMS, CMS–(NH)_3_–COOH, CMS–(NH)_3_–COOH@DOX, CMS–(NH)_3_–COOH@GB, and CMS–(NH)_3_–COOH@DOX@GB was evaluated by an MTT assay. Briefly, A549 cells were seeded in 96-well plates (8 × 10^3^ cells/well) and incubated in a 100 μL DMEM solution in a 5% CO_2_ atmosphere at 37 °C for 24 h. Then, different concentrations of the CMS, CMS–(NH)_3_–COOH, CMS–(NH)_3_–COOH@DOX, CMS–(NH)_3_–COOH@GB, and CMS–(NH)_3_–COOH@DOX@GB solutions were added to the plate and incubated for a further 48 h. Twenty microliters of the MTT solution (5 mg/mL) was added to each well and incubated for another 4 h. Afterwards, the medium was removed and replaced with 150 μL DMSO. After 10 min of incubation under gentle shaking, the optical-density (OD) values of the alive cells in the plate were provided from a microplate reader model 550 (BiO–Rad, USA) at 490 nm. The final data were determined by the mean value of 8 replicates for each sample.

### 2.5. In Vitro Drug Release

The release experiments of DOX and GB were taken in buffer solutions of pH 7.4 and 5.0, respectively, by heating the same volume of the buffer solution (50 mL) up to 37 °C in a water-bath environment. The nanoparticles (10 mg) were placed in 8000–14,000 MWCO (Molecular Weight Cut Off) dialysis bags (Beijing, China) with 4 mL of the buffer solution and stirred at 400 rpm. Immediately, 3 mL aliquots were taken, and another 3 mL of the fresh buffer solution was added at the same time. The absorbance of the 3 mL aliquot was measured at an interval of 480 and 290 nm wavelengths, respectively, using an ultraviolet spectrophotometer UV-6100s. The accumulative release rate was calculated as follows [31]:(1)$$C_{C}=C_{t}+\frac{v}{V}\sum_{0}^{t-1}C_{t}$$ where *C_c_* is the real concentration of DOX and GB released at time *t*; *C_t_* is the apparent concentration measured by UV-vis spectrometry of the release fluid sample at time *t*; *ν* is the sample volume taken at predetermined time intervals; and *V* is the total volume of the release fluid.

## 3. Results and Discussion

### 3.1. Structure and Characteristics of Materials

#### 3.1.1. SEM and TEM Analysis

Figure 1 and Figure 2 are the CMS, CMS–(NH)_3_, and CMS–(NH)_3_–COOH of the scanning electron microscopy and transmission electron microscopy images, respectively. The materials are similar to the coral-like microparticles composed of many nanoparticles held together (see Figure 1). Figure 2 shows the TEM images of samples of CMS, CMS–(NH)_3_, and CMS–(NH)_3_–COOH at different magnifications, indicating that the CMS before and after surface modification possesses an open-framework structure. The mesopore structure of CMS–(NH)_3_ and CMS–(NH)_3_–COOH could not be clearly observed (Figure 2D–I), which indicated the successful coating of the amino and carboxyl groups on the CMS. 

#### 3.1.2. FTIR Analysis

Figure 3 shows the FT-IR spectra of all samples. Figure 3A shows that the nonpurified CMS had no obvious characteristic peaks of PEG–PPG–PEG at 2891 and 2942 cm^−1^ (stretching vibration peaks of –CH_2_) and had the shear vibration peak of 1467 cm^−1^ of C–H [32]. Following template removal, its characteristic peaks disappeared (Figure 3A (b)), which confirmed an effective purification process [33]. However, Figure 3B shows obvious stretching vibration peaks of –CH_2_. The stretching vibration peaks of Si–O–Si appeared at 1085 and 1212 cm^−1^, and its asymmetry vibration peaks appeared at 805 cm^−1^. Si–OH had a symmetrical stretching vibration peak at 964 cm^−1^. The stretching vibration peaks of the Si–O bond appeared at 459 cm^−1^. As for AEPTMS, the bands at 2833 and 2941 cm^−1^ were stretching vibration peaks of –CH_2_; the other bands at 3431 and 3278 cm^−1^ are the NH stretching vibration peaks of the primary and secondary amines, and the shear vibration peak at 1471 cm^−1^ is –CH_2_. However, in Figure 3A (d,e) and Figure 3B (d–f), there is no obvious peak at 964 cm^−1^. This is because the head group on the AEPTM Sinteracted with the Si–OH with the hydrogen bond. In Figure 3A (d,e), the peak of 1562 cm^−1^ was the stretching vibration peak of the –NH_2_ bending of AEPTMS, indicating that the NH_2_ groups coupled to the CMS surface. The characteristic absorption peaks of –COOH were at 1721 cm^−1^. The peaks at 1561.83 and 1638.36 cm^−1^ corresponded to amide I (mainly N–H) and amide II (mainly C=O stretch) of the –NH_2_ on the AEPTMS interacting with the succinic anhydride, indicating the formation of CMS–(NH)_3_–COOH. Gobind et al. [34] and Shen et al. [35] thoroughly analyzed the FT-IR spectra of DOX and GB. After being loaded with DOX and GB, the characteristic absorption peaks of the benzene ring appeared between 1400 and 1600 cm^−1^, successfully manifesting DOX and GB grafted on the CMS surface. The peak at 2360 cm^−1^ is the characteristic peak of carbon dioxide that was caused by the system of FT-IR spectra.

#### 3.1.3. XRD Analysis

The phases of the samples (CMS, CMS–(NH)_3_ and CMS–(NH)_3_–COOH) were inspected by XRD (Figure 4). When the template of the mesoporous silica was removed, one distinct broad peak appeared at 2θ = 15–23°, which corresponded to a typical amorphous silica phase.

#### 3.1.4. XPS Analysis

To better evidence the amine and carboxyl successfully grafted on CMS, XPS measurements were carried out on CMS, CMS–(NH)_3_, and CMS–(NH)_3_–COOH. Through the surveyed spectra in Figure 5 and the data in Table 1, the five peaks of the XPS spectra in Chart A corresponding to silicon signals at 544.18, 410.18, 300.18, 155.08, and 110.18 eV represent the binding energies for Si2s, Si2p, O1s, N1s, and C1s, respectively. The C1s spectrum was divided into six different peaks attributed to binding energies for O–C=O (288.48 eV), C=O (287.78 eV), C–O (286.38 eV), C–N (285.58 eV), C–C (284.78 eV), and C–Si (C–Si–O) (283.74 eV), as shown in Figure 5B. The peak of C–Si (C–Si–O) reveals the presence of the AEPTMS that grafted onto CMS, and the presence of O–C=O and Si–C (O–Si–C) implies that the amine successfully transformed into carboxyl. The N1s spectrum was divided into six different peaks attributed to the binding energies for N–H (401.05 eV), O=C–N (399.9 eV), and C–N (C–N–C) (399.1 eV), as displayed in Figure 5C. The peaks of N–H and C–N (C–N–C) suggest the presence of multiamine chains, and only the tail amino carboxylated by succinic anhydride and the presence of C=O further confirmed it (see Figure 5E). The peak for O=C–N reveals the presence of acylamino.

#### 3.1.5. Nitrogen-Adsorption–Desorption Analysis

The N_2_ adsorption/desorption isotherms, BJH [36] pore size, and pore-volume distributions of CMS and CMS–(NH)_3_–COOH are shown in Figure 6. Figure 6A shows two samples of typical-type IV adsorption isotherms with H2 hysteresis loops in the IUPAC classification [37], indicating the presence of large amounts of mesopores. The peak pore diameter was evaluated to be about 20.12 nm (see Figure 6B). Moreover, the gradually decreased pore size of CMS–(NH)_3_–COOH confirmed the functionalized carboxyl. The average pore volume and surface area were estimated via the BET [38] method (see Table 2). Table 2 shows that, after modification, the pore size, specific surface area, and pore volume of CMS were decreased, which indicated part carboxyl chains in mesoporous channels. The decrease of specific surface areas is due to the presence of pendant organic chains covalently bonded to the inorganic network, partially blocking the entrance of nitrogen molecules [39], which indicates that organic groups were fastened in the mesopore.

#### 3.1.6. Zeta-Potential Analysis

In addition, the successful surface amine and carboxyl functionalization was confirmed through the zeta-potential measurement of different pH values (pH 5–8). As shown in Figure 7, the zeta potential of CMS was −24.4 mV, showing the existence of surface –OH. However, this increased to 5.97 mV (CMS–(NH)_3_) after being modified with positively charged amine groups. After the modification of succinic anhydride, the zeta potential decreased from 5.97 to −26.3 mV (CMS–(NH)_3_–COOH) for the strong negative charge of carboxyl groups rooting in SAD. Generally, nanoparticles are considered to be colloidally stable if the zeta-potential value is below −25 mV [40]. The functionalized CSM has zeta potential values in the pH range of 5–8, as following Figure 7B, indicating that the functionalized CSM is stabile in the pH range of 5–8. As a result, the introduction of SAD could enhance the stability of the CMS drug-delivery system, which is of critical importance for its therapeutic application [40].

### 3.2. Loading Analysis of DOX and GB

Five percent phosphate buffer saline was chosen as a solvent to load DOX for the hydrophilia of DOX, while DCM was applied for hydrophobic GB. According to Section 2.5, the calculated contemporary loading of DOX and GB were 25.31% and 44%, respectively, after the multifunctional groups were grafted. Based on our investigation, we used large-pore mesoporous silica as the carrier to simultaneously load DOX and GB and to achieve a high loading capacity for the first time. The previous works in Table 3 show that CMS–(NH)_3_–COOH had the highest loading for DOX and GB. The abundant polyamine chain and carboxyl of CMS–(NH)_3_–COOH made it easily react with the amine and hydroxyl of DOX and the amine of GB, and the pore diameter of the material itself (18.345 nm) is beneficial to loading DOX and GB.

## 4. In Vitro Cytotoxicity

The cytotoxicity of blank and DOX- and GB-loaded microparticles toward A549 cells was assessed using a 3-(4,5-dimethyl-2-thiazolyl)-2,5-diphenyl-2-H-tetrazolium bromide (MTT) assay. As can be seen in Figure 8A, after 48 h incubation, CMS and drugfree CMS–(NH)_3_–COOH showed no significant cytotoxicity on A549 cells at different concentration (125, 250, 500, and 1000 µg·mL^−1^), indicating both CMS and CMS–(NH)_3_–COOH microparticles were very biocompatible and could be accepted in vivo. Nanoparticles are biocompatible at a concentration of up to 2000 µg∙mL^−1^ [41]. The inhibition rate also increased with increasing concentration, as expected, manifesting a dose-dependent cytotoxic effect (see Figure 8B–D). The inhibition rates of CMS–(NH)_3_–COOH@GB and CMS–(NH)_3_–COOH@DOX were lower than those of GB and DOX (see Figure 8B,C).

Viability studies also demonstrated that CMS–(NH)_3_–COOH@GB greatly increases the cell inhibition rate at concentrations as high as 644.08 μg∙mL^−1^, which is comparable to the cytotoxic effect of free GB (IC50, 154.52 μg∙mL^−1^). The independent CMS–(NH)_3_–COOH@GB and CMS–(NH)_3_–COOH@DOX had a lower efficacy for GB and DOX than the free GB and DOX (see Table 4). The half-maximum inhibitory concentration (IC50) value of CMS–(NH)_3_–COOH@ DOX@GB against A549 cells was calculated to be approx. 51.27 μg∙mL^−1^, suggesting a fairly high therapeutic effectiveness and better synergies for DOX and GB. However, we expect that the CMS–(NH)_3_–COOH@DOX@GB selectively delivers the cytotoxic agent to acidic cancerous cells.

## 5. Release of DOX and GB

Figure 9A–F shows the accumulative release curve of DOX and GB from CMS–(NH)_3_–COOH@DOX@GB, CMS–(NH)_3_–COOH@DOX, and CMS–(NH)_3_–COOH@GB in different release fluids with different pH values (pH 5.0 and 7.4). At pH 7.4, as shown in Figure 9A, the accumulative release rates of DOX and GB were 8.45% and 4.83%, respectively, indicating that mesoporous materials increased GB solubility. We could obtain a result in which the GB first started to release and reached 0.25%, but the DOX release rate was zero, and the release rate of GB exceeded that of DOX within 3 h (see Figure 9A,a). After 3 h, the release of DOX gradually increased and surpassed GB. After 40 h, the release curves of DOX and GB tended to be gentle (see Figure 9A). At pH 5.0, the accumulative release rates of DOX and GB were 32.03% and 13.66%, respectively. The hydrogen bond reaction between the –COOH of the modified materials and the –OH and amine of drug molecule increased GB solubility and reached the acid solution better, as shown in Figure 9D. It is worth noting that DOX and GB were released simultaneously, with approximate release rates of 0.084% and 0.08%, respectively. The release of DOX was continuously higher than that of GB (see Figure 9B,b). At pH 5.0 and 7.4, the general releasing trend of DOX was higher than GB, which related to the hydrophilia of DOX and the hydrophobia of GB. Figure 9E,F are the release curves of CMS–(NH)_3_–COOH@GB and CMS–(NH)_3_–COOH@DOX, respectively. The results show that, in the GB systems, the release rate of GB is higher than the system containing DOX and GB at pH 5.0. However, in the DOX systems, the release rate of DOX is lower than the system containing DOX and GB at pH 5.0 and at pH 7.4, indicating that the release rate of DOX is enhanced in the system containing both drugs; however, the GB is weakened. The higher release rate of DOX and GB in the acid solution manifested the pH response property of CMS–(NH)_3_–COOH (see Figure 9C,D). This shows that the carboxyl of the polyamine chain is more conducive to controlled drug release.

A pH-controlled “open–close” mechanism (see Scheme 2) for the release of DOX and GB from CMS–(NH)_3_–COOH@DOX@GB could explain the phenomenon. With previous studies of different “open–close” mechanisms [42] at low pH (pH 5.0), the hydrogen–bonding interactions between the electronegative CMS–(NH)_3_–COOH and abundant positive charges in a weak acid solution could lead to the Coulomb attraction of head–head of CMS–(NH)_3_–COOH greatly weakening or disappearing. However, multiamine chains could be protonated and could contribute to the generation of Coulombic repulsion; thus, the “open gate” of CMS–(NH)_3_–COOH emerges. At a high pH (pH 7.4), the Coulomb repulsion of head–head of CMS–(NH)_3_–COOH would exist between the electronegative CMS–(NH)_3_–COOH and abundant negative charges in a weak basic solution. However, the Coulombic attraction between positively charged multiamine chains and CMS–(NH)_3_–COOH in a weak basic solution leads to the appearance of the “close gate” of CMS–(NH)_3_–COOH. Thus, the release of DOX and GB at a low pH (pH 5.0) was faster than at a high pH (pH 7.4, see Figure 9C,D).

## 6. Conclusions

In summary, we obtained a dual drug-delivery system through developing a pH-sensitive carboxyl-functionalized large-pore CMS–(NH)_3_–COOH to codeliver a synergistic gefitinib/DOX combination for A549 cells. Their release behavior could be separately controlled in response to different pH values. We used a pH–controlled “open–close” mechanism (see Scheme 2), which better explained the release behaviors of DOX and GB from CMS–(NH)_3_–COOH@DOX@GB. According to a cytotoxicity analysis, this double-drug system with microparticles and large pores makes GB and DOX have a good long-term synergistic therapeutic effect. This drug-delivery system is a valuable pH response strategy for the delivery of anticancer drugs, and it provides a potential strategy to overcome drug resistance.

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
