# Peer review of "Functionalized Large-Pore Mesoporous Silica Microparticles for Gefitinib and Doxorubicin Codelivery"

_materials, 2019, doi:10.3390/ma12050766_

Reviewer 1 Report

The authors report the synthesis of mesoporous silica microparticles using a triblock polymer as structure directing template, for the controlled release of doxoribicin and gefitinib. The issues treated in this paper have relevance in drug delivery for cancer therapy. I believe that the paper can be accepted for publication in this Journal after addressing some issues.

Firstly, the English language and grammar must be revised and some too short sentences should be avoided.

The stability of the functionalized CSM designed as drug delivery systems should better investigated by reporting the zeta potential values in the pH range of 5-8.  

The release of DOX and GB was evaluated only in the system containing both drugs; in order to investigate if any possible interactions between the two drugs could modify the drugs release, this study must be performed also for the two systems containing DOX or GB. Moreover, in my opinion this release should be investigated also at basic pH.

Author Response

Point1: Firstly, the English language and grammar must be revised and some too short sentences should be avoided.

Response 1: The article has has undergone English language editing by MDPI. The text has been checked for correct use of grammar and common technical terms, and edited to a level suitable for reporting research in a scholarly journal.

Point2: The stability of the functionalized CSM designed as drug delivery systems should better investigated by reporting the zeta potential values in the pH range of 5-8.  

Response 1: The functionalized CSM has been done the zeta potential values in the pH range of 5-8, as following figure1, indicating that the functionalized CSM is stability in the pH range of 5-8.

Figure1. the zeta potential values of functionalized CSM in the pH range of 5-8

Point3:The release of DOX and GB was evaluated only in the system containing both drugs; in order to investigate if any possible interactions between the two drugs could modify the drugs release, this study must be performed also for the two systems containing DOX or GB. Moreover, in my opinion this release should be investigated also at basic pH.

Response 3: The figure2A and figure2B are the release curves of CMS–(NH)3–COOH@GB and CMS–(NH)3–COOH@DOX. The results show that in the GB systems the release rate of GB is higher than the system containing DOX and GB at pH5.0. But in the DOX systems the release rate of DOX is lower than the system containing DOX and GB at at pH5.0 and at pH 7.4, indicated that the release rate of DOX is enhanced in the system containing both drugs, however the GB is weaken.. The figure2C is the release curve of CMS–(NH)3–COOH@DOX@GB at basic pH 8, the result shows that the release rates of DOX and GB get lower at basic environment.

 Figure2. The release curves of CMS–(NH)3–COOH@GB(A), CMS–(NH)3–COOH@DOX(B) and CMS–(NH)3–COOH@DOX@GB(C)

Reviewer 2 Report

The authors proposed an interesting research subject addressed through wide modern complementary testing methods, which could lead to qualitative and valuable results. However, the manuscript lacks a logical and chronological presentation of methods and results sections, along with extensive speculative interpretation of data. We outline below the respective issues inconsistent with this article publication:

1. Overall the article is incomprehensible presented in a poor English manner, with complicated structure and tangled chapters.

2.  The whole Materials and Methods and Results and Discussion sections should be redesigned, reorganized and completed (e.g sample preparation for CMS-(NH3)-COOH@GB is not provided). For sample preparation, full argumentation of involved parameters should be described or pointed to a previously reported paper (e.g. different stirring rpm values or inexistent, different temperatures presented either in K/°C scale and drying/stirring times, etc).

3.  Expressions used on lines 156 and 171 do not correlate with the required scientific writing.

4.  Regarding the results section, all samples should have been characterized by all methods (e.g. see BET, XRD and FT-IR, XPS with different number of investigated samples). For SEM micrograph in figure 1A, authorship for coralline structure (upper right corner) is not provided.

5.  The authors provide speculative and incomplete SEM and TEM discussion (e.g. samples do not present open channels – at most SEM results depict particle agglomerations and TEM ones the spaces between particles).

6.  If the figure 5 caption is correct, how can 2 spectra of the same material be different? (e.g. sample A-e and B-a).

7.  If the same amount of DOX and GB are added why is the coordinate axes different for CMS-(NH3)-COOH@GB and CMS-(NH3)-COOH@DOX in figure 8? On line 167 “different concentrations of CMS [..]” should have been clearly presented and related with the results in figure 8 and table 4.

8.  In Conclusions section the authors mention the MSN-(NH3)-COOH system for the first time. Should we assume this is related to your previous study? Also, presented results do not conduct to the conclusion of a “new pH-controlled “open-close” mechanism”.

Author Response

Point1: Overall the article is incomprehensible presented in a poor English manner, with complicated structure and tangled chapters.

Response 1: The article has undergone English language editing by MDPI. The text has been checked for correct use of grammar and common technical terms, and edited to a level suitable for reporting research in a scholarly journal.

Point  The whole Materials and Methods and Results and Discussion sections should be redesigned, reorganized and completed (e.g sample preparation for CMS-(NH3)-COOH@GB is not provided). For sample preparation, full argumentation of involved parameters should be described or pointed to a previously reported paper (e.g. different stirring rpm values or inexistent, different temperatures presented either in K/°C scale and drying/stirring times, etc).

Response2: The whole Materials and Methods and Results and Discussion sections of article have been redesigned and reorganized (see revised Manuscript). At the same time, we checked and supplement all of the involved parameters in our revised manuscript.

For example: Four grams of triblock poly (ethylene glycol)-b-poly (propylene glycol)-b-(poly(ethylene glycol) (PEG-b-PPG-b-PEG), MW = 8400, Aladdin, ShanghaiChina) and 20 ml of hydrochloric acid (36~38%, Chongqing, China) were added into 120 ml deionized water. This was stirred 20 min until the solution was clarified at room temperature; we then added a 6 g sodium silicate (Komeo, Tianjin, Chian) solution (10.5 g sodium silicate dissolved in 10.5 g deionized water). After stirring for 4 h, 15 ml anhydrous ethanol was added to the rest of the 15 g sodium silicate solution, dropwise adding the mixed solution at a rate of 0.3 ml/s. Next, it was washed at constant pressure in a split funnel with 5 ml deionized water; we obtained a pale-blue solution after stirring for 24 h at 1200 rpm. The above mixed solution was poured into the reaction kettle with the polytetrafluoroethylene gallbladder hydrothermal for 24 h at 120  with a heating rate of 5/min. The mixture was obtained by vacuum-suction filter, washed three times with deionized water and anhydrous ethanol, and dried overnight in a drum wind-drying oven at 60 . The obtained solid white powder was calcined at 6 h in a high-temperature tubular-resistance furnace of 550 on air with a temperature of 5/ min; this was named CMS.

About sample preparation for CMS-(NH3)-COOH@GB: Gefitinib (GB, Aladdin, Shanghai, China) was loaded: 30mg GB was dissolved in 30ml dichloromethane (DCM) and 60mg CMS-(NH)3-COOH were added. They were stirred at room temperature, 400 rpm for 24 h keeping dark condition. The loaded-samples (named CMS-(NH)3-COOH@GB) were collected after centrifuged at 8000 rpm for 10 min, washed the sediment to clarification with DCM and dried in air dry oven at 50 °C for 24 h. The following steps are the same as above except the absorbance was measured at 252 nm.

Point 3.  Expressions used on lines 156 and 171 do not correlate with the required scientific writing.

Response 3: About expressions used on lines 156 and 171 have been deleted.

Point 4.  Regarding the results section, all samples should have been characterized by all methods (e.g. see BET, XRD and FT-IR, XPS with different number of investigated samples). For SEM micrograph in figure 1A, authorship for coralline structure (upper right corner) is not provided.

Response 4: All samples have been characterized by all methods. The figure1 shows N2 adsorption–desorption curve, and pore distribution of CMS, CMS–(NH)3 and CMS–(NH)3–COOH. The figure2 shows the Fourier-transform infrared (FTIR) spectra of all samples. The figure3 shows the X-ray photoelectron spectroscopy (XPS) survey spectra of all samples.

Figure1. Brunauer–Emmett–Teller (BET) N2 adsorption–desorption curve, and pore distribution of CMS, CMS–(NH)3 and CMS–(NH)3–COOH.

Figure2. Fourier-transform infrared (FTIR) spectra of A: (a) CMS/PEG-PPG-PEG, (b) CMS, (c) AEPTMS, (d) CMS–(NH)3, (e) CMS–(NH)3–COOH; B: (a) CMS–(NH)3–COOH. (b) DOX, (c) GB, (d) CMS–(NH)3–COOH@DOX, (e) CMS–(NH)3–COOH@GB, (f) CMS–(NH)3–COOH@DOX@GB.

Figure3. X-ray photoelectron spectroscopy (XPS) survey spectra for (A) CMS, CMS–(NH)3 and CMS–(NH)3–COOH.

Point 5.  The authors provide speculative and incomplete SEM and TEM discussion (e.g. samples do not present open channels – at most SEM results depict particle agglomerations and TEM ones the spaces between particles).

Response 5: About the expression of SEM and TEM, they have been corrected in the original manuscript, as following: the CMS of the scanning electron microscopy and transmission electron microscopy, respectively. The materials are similar to coral-like microparticles composed of many nanoparticles held together (see Figure 1A and 1B),indicating that the agglomeration between particles is serious. Figure 2 shows the TEM images of CMS at different magnifications, indicating that the CMS possesses an unambiguous channel structure between particles.

Point 6.  If the figure 5 caption is correct, how can 2 spectra of the same material be different? (e.g. sample A-e and B-a).

Response 6: About the figure 5, we used two different instruments in figure5 A and 5B (nowadays is figure3A and B) before and contributed to be different from 2 spectra of the same material. We retested all the samples with the same instrument in our revised manuscript. The figure 4 shows the Fourier-transform infrared (FTIR) spectra of all samples.

Figure4. Fourier-transform infrared (FTIR) spectra of A: (a) CMS/PEG-PPG-PEG, (b) CMS, (c) AEPTMS, (d) CMS–(NH)3, (e) CMS–(NH)3–COOH; B: (a) CMS–(NH)3–COOH. (b) DOX, (c) GB, (d) CMS–(NH)3–COOH@DOX, (e) CMS–(NH)3–COOH@GB, (f) CMS–(NH)3–COOH@DOX@GB.

Point 7.  If the same amount of DOX and GB are added why is the coordinate axes different for CMS-(NH3)-COOH@GB and CMS-(NH3)-COOH@DOX in figure 8? On line 167 “different concentrations of CMS [..]” should have been clearly presented and related with the results in figure 8 and table 4.

Response 7: About the sentence “with equivalent concentrations of GB(B) and DOX(C) for 48 h.”, we have no clear expression. That’s mean that concentration of single GB is same with concentration of GB in CMS–(NH)3–COOH@GB and the concentration of single DOX is same with concentration of DOX in CMS–(NH)3–COOH@DOX. I have corrected this expression. The different concentrations of CMS, CMS–(NH)3–COOH, CMS–(NH)3–COOH, DOX,GB,CMS–(NH)3–COOH@DOX, CMS–(NH)3–COOH@GB and CMS–(NH)3–COOH@DOX@GB as following table:

Sample

End   concentration(mg/ml)

inhibition   ratio(%)

IC50 value

CMS

2000

21.35±0.64

1000

4.55±5.96

500

4.30±5.37

250

0

125

0

CMS-(NH)3-COOH

2000

23.00±8.13

1000

3.18±2.18

500

1.66±4.14

250

0

125

0

GB

200

67.18±6.64

150

44.79±10.49

100

33.19±6.49

154.52μg/ml

50

13.58±0.96

25

4.79±3.17

DOX

2

73.29±1.38

1

64.20±8.96

0.8

49.85±8.08

0.80μg/ml

0.4

38.92±7.03

0.2

33.16±5.16

CMS-(NH)3COOH@DOX

10(Including DOX 2mg/ml)

14.93±5.76

5(Including DOX 1mg/ml)

0

4(Including DOX 0.8mg/ml)

0

2(Including DOX 0.4mg /ml)

2.97±2.79

1(Including DOX 0.2mg/ml)

4.87±4.60

CMS-(NH)3-COOH@GB

660(Including GB 200mg/ml)

54.97±8.54

495(Including GB 150mg/ml)

30.76±13.82

330(Including GB 100mg/ml)

16.22±2.83

644.08μg/ml

165(Including GB 50mg/ml)

13.63±5.01

87.5(Including GB 25mg/ml)

3.65±0.72

CMS-(NH)3COOH@DOX@GB

80

73.79±12.91

60

58.97±8.84

30

36.14±1.34

51.27μg/ml

15

13.21±3.66

7.5

8.50±2.66

Point 8.  In Conclusions section the authors mention the MSN-(NH3)-COOH system for the first time. Should we assume this is related to your previous study? Also, presented results do not conduct to the conclusion of a “new pH-controlled “open-close” mechanism”.

Response 8: About MSN-(NH3)-COOH system, it is not related to my previous study. It was belong to errors of expression and writing that some unapt expressions have been corrected.

   About the conclusion of a “new pH-controlled “open-close” mechanism”, presented results did not indeed conduct to the conclusion, the conclusions of article has been amended as following:

In summary, we obtained a dual drug-delivery system through developing a pH-sensitive carboxyl-functionalized large-pore CMS(NH)3COOH to codeliver a synergistic gefitinib/DOX combination for A549 cells. Their release behavior could be separately controlled in response to different pH values. We used a pH–controlled “open–close” mechanism (see Scheme 2), which better explained the release behaviors of DOX and GB from CMS–(NH)3–COOH@DOX@GB. According to cytotoxicity analysis, this double-drug system with microparticles and large pores makes GB and DOX have a good long-term synergistic therapeutic effect. This drug-delivery system is a valuable pH response strategy for the delivery of anticancer drugs, and it provides a potential strategy to overcome drug resistance.

Reviewer 3 Report

The manuscript is well written and the details of the experimental sections were described properly. The manuscript is acceptable in the current form.

Author Response

The manuscript is well written and the details of the experimental sections were described properly. The manuscript is acceptable in the current form.

Response: Thank you very much for your affirmation and recognition of our work. I wish you a higher career!

Round  2

Reviewer 2 Report

The authors proposed an interesting research subject addressed through wide modern complementary testing methods, which could lead to qualitative and valuable results. After the first revision the manuscript was amended according to a logical and chronological presentation of methods and results sections. We outline below the respective issues that were not addressed:

Point 2: On section 2.3 the third paragraph is identic with the second one.

Point 4: SEM and TEM analysis do not illustrate the results for all samples: CMS, CMS-(NH)3, CMS-(NH)3-COOH, CMS-(NH)3-COOH@DOX, CMS-(NH)3-COOH@GB, CMS-(NH)3-COOH@DOX@GB. Also XRD analysis should present the same number of analyzed samples as the FT-IR one.

Point 5: The interpretation of SEM and TEM results is still incomplete and may be due to the fact that not all samples are displayed. Also, the authors use in the new manuscript version, the syntax “unambiguous channel structure” related to TEM results, meanwhile for the SEM ones they depicted a “serious agglomeration between particles”. Previously we stated that there was no channel structure, so what is the term “unambiguous” used for?

Point 7:  We understand now that the “different concentrations” are related with the in vitro preparation process. But why aren`t they investigated at the same concentration? This analysis leads to a confusion in figure 8 (C) which can be correlated with a lack of results, given that the table provided in the reviewer`s response is not completely integrated in the manuscript.

Point 9: On table 2 in the new version of the manuscript, authors inserted the missing sample, but however the numerical results do not match. They expose an inversion of data between samples CMS-(NH)3 and CMS-(NH)3-COOH and provide new results for the sample CMS-(NH)3-COOH, which was originally included in the manuscript.

Author Response

Response to Reviewer 2 Comments

Point 2: On section 2.3 the third paragraph is identic with the second one. 

Response 2: Thanks for pointing out this. We added the second paragraph for answer the question point 2 "sample preparation for CMS-(NH3)-COOH@GB is not provided" that you asked in last comments, but we didn't express clearly obviously. So we made the change of the third paragraph in the revised manuscript this time. As following:

GB and DOX was co-loaded: 30 mg GB was dissolved in 30 ml dichloromethane (DCM), and 31 mg CMS–(NH)3–COOH@DOX was added. The following steps were the same as above, and the loaded samples was named CMS–(NH)3–COOH@DOX@GB . To calculate the DOX and GB loading rate respectively, we used ultraviolet spectrophotometer UV-6100s at 480 nm and 252 nm and took the average after repeating three times to measure absorbance respectively.

Point 4: SEM and TEM analysis do not illustrate the results for all samples: CMS, CMS-(NH)3, CMS-(NH)3-COOH, CMS-(NH)3-COOH@DOX, CMS-(NH)3-COOH@GB, CMS-(NH)3-COOH@DOX@GB. Also XRD analysis should present the same number of analyzed samples as the FT-IR one.

Response 4: We supplemented some SEM and TEM images in figure 1 and figure 2 and made the change of 3.1.1, as following. But combined with the XRD diagram in Figure 4, three patterns (CMS, CMS-(NH)3 and CMS-(NH)3-COOH) all showed one distinct broad peak appeared at 2θ = 15-23°, which corresponded to a typical amorphous silica phase. In addition, as pointed out in the references that we referred ([1]Jingjing WYajing JYiran Sh.  Materials, 2018, 11, 2041. [2]Malfait B , Correia N T , Mussi A , et al. Microporous and Mesoporous Materials, 2018. [3] Gao Q, Xu Y, Wu D, et al. Langmuir the Acs Journal of Surfaces & Colloids, 2010, 26(22):17133.), researchers usually evaluate the morphology and depict particle agglomerations of materials before and after surface modification by TEM and SEM respectively. So we only provided SEM, TEM and XRD analysis of samples before and after surface modification but not all samples. 

3.1.1. SEM and TEM Analysis

Figures 1 and 2 are the CMS, CMS–(NH)3 and CMS–(NH)3–COOH of the scanning electron microscopy and transmission electron microscopy images, respectively. The materials are similar to coral-like microparticles composed of many nanoparticles held together (see Figure 1). Figure 2 shows the TEM images of samples of CMS, CMS–(NH)3 and CMS–(NH)3–COOH at different magnifications, indicating that the CMS before and after surface modification possesses open-framework structure. The mesopore structure of the CMS–(NH)3 and CMS–(NH)3–COOH could not be clearly observed (Figure 2D,E,F,G,H,I), which indicated the successful coating of amino and carboxyl groups on the CMS.

Figure 1. Scanning electron microscopy images of (A, B) CMS, (C, D) CMS–(NH)3 and (E, F) CMS–(NH)3–COOH.

Figure 2. Transmission electron microscopy (TEM) images of CMS (A, B, C), CMS–(NH)3 (D, E, F) and CMS–(NH)3–COOH(G,H,E).

Point 5: The interpretation of SEM and TEM results is still incomplete and may be due to the fact that not all samples are displayed. Also, the authors use in the new manuscript version, the syntax “unambiguous channel structure” related to TEM results, meanwhile for the SEM ones they depicted a “serious agglomeration between particles”. Previously we stated that there was no channel structure, so what is the term “unambiguous” used for?

Response 5: We are so sorry that we made this mistake for misunderstood you meant. We supplemented some SEM and TEM images in figure 1 and figure 2 and made the change of 3.1.1 in the third manuscript version. Same as the response 4 above.

Point 7:  We understand now that the “different concentrations” are related with the in vitro preparation process. But why aren`t they investigated at the same concentration? This analysis leads to a confusion in figure 8 (C) which can be correlated with a lack of results, given that the table provided in the reviewer`s response is not completely integrated in the manuscript.

Response 7: When we designed this experiment, we focused on the comparison between drug carrier samples and pure drugs. In the experiment, higher concentration was not considered since the toxicity of DOX to cells was significantly different from that of drug carrier materials when the concentration was very low.

We integrated all the results in the table that we provided to the reviewer before into the figure 8 of this manuscript, and reorganized the description of the figure to try to correct the original confusion. As following:

4. In Vitro Cytotoxicity

The cytotoxicity of blank, and DOX- and GB-loaded microparticles toward A549 cells was assessed using a 3-(4, 5-dimethyl-2-thiazolyl)-2, 5-diphenyl-2-H-tetrazolium bromide (MTT) assay. As can be seen in Figure 8A, after 48 h incubation, CMS and drugfree CMS–(NH)3–COOH showed no significant cytotoxicity on A549 cells at different concentration (125, 250, 500 and 1000 µg∙mL-1) , indicating both CMS and CMS–(NH)3–COOH microparticles were well-biocompatible and could be accepted in vivo. Nanoparticles are biocompatible at a concentration of up to 2000 µg∙mL-1[42]. The inhibition rate also increased with increasing concentration, as expected, manifesting a dose- -dependent cytotoxic effect (see Figure 8B, 8C and 8D). The inhibition rates of CMS–(NH)3–COOH@GB and CMS–(NH)3–COOH@DOX were lower than those of GB and DOX (see Figure 8B and 8C ).

Figure8. In vitro cytotoxicity of CMS, CMS–(NH)3–COOH, DOX, GB, CMS–(NH)3–COOH@GB, CMS–(NH)3–COOH@DOX, and CMS–(NH)3–COOH@DOX@GB for treated A549 cells

Point 9: On table 2 in the new version of the manuscript, authors inserted the missing sample, but however the numerical results do not match. They expose an inversion of data between samples CMS-(NH)3 and CMS-(NH)3-COOH and provide new results for the sample CMS-(NH)3-COOH, which was originally included in the manuscript.

Response 9: Thanks for pointing out this.  We didn't provide the results of sample  CMS-(NH)3 in our original manuscript, but the results of CMS-(NH)3 was mistakenly written into table 2 as the results of CMS-(NH)3-COOH. We didn't find this mistake until inserted the missing sample, so provided new results for the sample CMS-(NH)3-COOH. Thank you very much for pointing out this mistake. We will be more rigorous in our future work. Thank you very much!

Dear professor, thank you for pointing out our shortcomings. Some questions exist in our studies for inconsiderateness. At the same time, since English is not our native language, there are many problems in both the writing of articles and the communication and understanding with reviewers. Sometimes, we cannot express our ideas clearly. We will try our best to improve our work in the future. Thank you very much!